# Diagnostic Utility of Genome-Wide DNA Methylation Analysis in Mendelian Neurodevelopmental Disorders

**DOI:** 10.3390/ijms21239303

**Published:** 2020-12-06

**Authors:** Sadegheh Haghshenas, Pratibha Bhai, Erfan Aref-Eshghi, Bekim Sadikovic

**Affiliations:** 1Department of Pathology and Laboratory Medicine, Western University, London, ON N6A 3K7, Canada; shaghsh@uwo.ca; 2Molecular Genetics Laboratory, Molecular Diagnostics Division, London Health Sciences Centre, London, ON N6A 5W9, Canada; pratibha.bhai@lhsc.on.ca; 3Division of Genomic Diagnostics, Children’s Hospital of Philadelphia, Philadelphia, PA 19104, USA; arefeshghe@chop.edu; 4Schulich School of Medicine and Dentistry, Western University, London, ON N6A 5C1, Canada

**Keywords:** epigenetics, DNA methylation, episignature, neurodevelopmental disorders, overgrowth with intellectual disability syndromes, constitutional disorders, machine learning, support vector machines, random forest

## Abstract

Mendelian neurodevelopmental disorders customarily present with complex and overlapping symptoms, complicating the clinical diagnosis. Individuals with a growing number of the so-called rare disorders exhibit unique, disorder-specific DNA methylation patterns, consequent to the underlying gene defects. Besides providing insights to the pathophysiology and molecular biology of these disorders, we can use these epigenetic patterns as functional biomarkers for the screening and diagnosis of these conditions. This review summarizes our current understanding of DNA methylation episignatures in rare disorders and describes the underlying technology and analytical approaches. We discuss the computational parameters, including statistical and machine learning methods, used for the screening and classification of genetic variants of uncertain clinical significance. Describing the rationale and principles applied to the specific computational models that are used to develop and adapt the DNA methylation episignatures for the diagnosis of rare disorders, we highlight the opportunities and challenges in this emerging branch of diagnostic medicine.

## 1. Chronology of Diagnostic Approaches for Neurodevelopmental Disorders

Rare genetic disorders, though individually rare, collectively constitute a large and heterogeneous group of diseases with a total estimated global prevalence of approximately 3.5–5.9% [1]. Rare genetic conditions pose a major health priority due to their chronic nature with long-term complications, often resulting in fatal outcomes with no genetic cause known for one-third of these conditions. Diagnosing a patient with a suspected rare genetic disorder is a challenging, long, and expensive process with a high failure rate. Neurodevelopmental disorders (NDDs) are a well-recognized class of rare genetic disorders, characterized by abnormal brain and central nervous system development because of genetic defects in genes controlling essential neurodevelopmental processes. Another well-studied group of rare genetic diseases is overgrowth/intellectual disability syndromes (OGIDs), with characteristic features including increased growth, macrocephaly, distinctive facial features, various degrees of learning difficulties and intellectual disability (ID), and increased susceptibility to cancer [2].

NDDs are a major health concern, as they affect >3% of children worldwide [3]. An accurate and timely diagnosis is critical to provide the best possible management and, sometimes, improve the overall disease outcome. Despite the current technical progress in sequencing technologies, the evaluation of the genetic causes of NDDs remains challenging because of genetic and phenotypic heterogeneity. For instance, patients with CHARGE and Kabuki syndromes have similar clinical features and are difficult to differentiate [4]. There is also an overlap between the characteristic features of Floating-Harbor syndrome (FLHS) and Rubinstein-Taybi syndrome 1 (RSTS1) because of the molecular interactions between their associated genes [4,5]. An additional challenge is phenotypic expressivity, where patients with the same genetic defects present with varying degrees of symptoms and phenotypes [6,7,8,9].

### 1.1. Conventional Genetic Testing Methods

Previously, G-banded karyotyping and FMR1 trinucleotide repeat analysis were the first-tier tests for patients with unexplained NDDs. However, the diagnostic yield in patients was low. The expected yield of FMR1 trinucleotide repeat analysis was 1%–5% in males [10], and the diagnostic yield of G-banded karyotype for patients with NDD developmental delay, intellectual disability, and autism spectrum disorder (ASD) was 5% [10,11]. With technological advancements and the growing need to make an accurate diagnosis in more patients, the implementation of chromosome microarray analysis (CMA) as part of the first-line evaluation for children with NDD achieved a diagnostic yield of 10–15% [12,13]. Gene panel, whole-exome, and whole-genome sequencing technologies (WES and WGS, respectively) have enhanced the ability to diagnose the NDDs. Of these, WES achieves a diagnostic rate of 30–53% for NDDs when used as a first-tier test, which more than doubles the rates achieved by CMA [13]. WGS is a comprehensive genetic test that improves the diagnostic rates even further (42-62%). It can offer a broader range of variant detection, including noncoding and regulatory regions, and has the ability to discover novel disease-associated genes, especially in cases with an ambiguous diagnosis [14,15,16,17,18]. Although WGS seems promising, its adoption as a clinical test is hampered by technical complexities, a high cost, and a lack of standardized clinical practice guidelines. The diagnostic flowchart of NDDs increased the percentage of patients receiving a confirmed molecular diagnosis [13,19]. Despite comprehensive genomic analyses, approximately half of NDD patients are left with an indefinite or inconclusive genetic diagnosis. Variants of uncertain significance (VUSs) in genes known to be associated with the disease often leave these patients and their families more confused and uncertain about the diagnosis. Sometimes, variant co-segregation studies within the family, in silico pathogenicity prediction tools, and experimental functional studies may guide the assessment of a VUS as being pathogenic or benign, but for most cases, this remains challenging in the clinic, leaving the diagnosis and management of these patients unclear. Other ambiguous genetic findings involve the identification of variants in noncoding parts of the genome, which further adds to the complexity. These existing gaps in achieving a successful diagnosis in unresolved cases led researchers to look for alternative diagnostic methods.

### 1.2. Epigenetics and Its Diagnostic Utility in NDDs

Epigenetics underlines heritable changes in gene expression without altering the underlying DNA sequence [20]. It has become increasingly clear that genes controlling epigenetic functions are significant contributors towards the underlying genetic basis of NDDs and OGIDs [2,21]. DNA methylation is the most common and well-understood epigenetic mechanism, which involves the addition of a methyl group to cytosine residues, and rare genetic variants have been reported in patients with unexplained NDDs in genes controlling the DNA methylation pattern [22,23,24]. Another well-studied epigenetic process is posttranslational modification of the histone tails [25,26]. DNA methylation and histone modification affect differential chromatin packaging, and they regulate access to specific areas of the genome. This, in turn, influences the ability of protein complexes of the transcriptional machinery to bind and interact with the underlying DNA sequence and regulate the gene expression. These processes, which are collectively called the “epigenetic machinery” [22], play a critical role in the cellular lineage determination during development, enabling phenotypic differentiation across the tissue types, despite sharing an identical germline DNA sequence.

In recent years, various methods have been developed to study genome-wide DNA methylation changes, which can be broadly classified as either next-generation sequencing (NGS)-based or array-based technologies. A plethora of genome-wide DNA methylation analysis methods like enzyme digestion, affinity enrichment, and sodium bisulfite conversion-based methods have been developed, but each one has its own advantages and limitations, which is beyond the scope of this article [27]. NGS platforms, however, allow a genome-wide investigation of DNA methylation profiles at a single base resolution level, but implementing it in a clinical setting is not yet feasible due to the increased cost required for the special infrastructure, data storage capacity, and bioinformatics support required to analyze and interpret the extensive amount of data generated by NGS. Microarray hybridization has become one of the most frequently used technologies for DNA methylation studies at the genome-wide level, as it allows a high-throughput DNA methylation analysis in a cost-effective manner, but has variable coverage based on the array design [28].

Unique genomic DNA methylation patterns, called “episignatures”, are promising alternatives to diagnose NDDs and OGIDs [20]. If the methylation change is observed at a large number of loci across the genome of patients with a confirmed diagnosis of the disorder compared to controls, the methylation data at the differentially methylated sites is called the episignature of that syndrome [29]. These episignatures are highly reliable, reproducible, and unique, and the evidence of their utility in diagnosing many rare genetic disorders is growing significantly. The computational process of establishing an episignature contains two principal sections: first, probe selection, which is the detection of CpG sites that are differentially methylated in patients compared to healthy controls, and second, the construction of a classifier using statistical and machine learning methods that can distinguish cases from controls using the selected probes (Figure 1). In Section 3, we will discuss different techniques and computational procedures used in developing classifiers. These recent developments paved the way for using epigenomics from a mere research concept to a clinical laboratory test, impacting diagnostic and therapeutic decisions in the clinic.

### 1.3. Syndromes with a Known Episignature

DNA methylation patterns in the mammalian genome are established at specific time points, mostly in early development. Errors in methylation due to a mutation in one of the genes involved in methylation regulation can lead to rare genetic syndromes, including NDDs [30,31,32]. Using episignatures in hereditary conditions was under investigation for many years, with reports describing differentially methylated regions in individuals with Down syndrome and Cornelia de Lange syndrome (CdLS) [33,34]. Since then, many other NDDs/OGIDs have been associated with episignatures. Our recent work on detecting syndrome-specific episignatures for 34 constitutional conditions enabled the use of the EpiSign analysis in clinical diagnostics [35] (Table 1). Many of these syndromes are associated with genes regulating the epigenetic machinery. An episignature for mental retardation, X-linked syndromic, Claes-Jensen-type (MRXSCJ) is associated with a histone H3 lysine 4 demethylase gene KDM5C [36]. Additionally, Sotos syndrome 1 (Sotos1), Tatton-Brown–Rahman syndrome (TBRS), and Weaver syndrome (WVS) are OGIDs associated with epigenetic regulatory genes [2,37,38,39]. Although many studies are focused on the study of syndromes resulting from mutations in chromatin regulatory genes, DNA methylation signatures are also being mapped in genetic conditions involving genes with functions unrelated to the epigenetic mechanisms, such as *SMS* and *UBE2A*, associated with mental retardation, X-linked, syndromic, Snyder-Robinson-type (MRXSSR) and mental retardation, and X-linked, syndromic, Nascimento-type (MRXSN), respectively [35]. Table 1 shows a current list of NDD syndromes for which a DNA methylation signature has been discovered. 

## 2. DNA Methylation Signatures, Concepts, and Principles

### 2.1. Comparison of Blood Tissue with Other Tissues for Detecting an Episignature

The most commonly used tissue for a DNA methylation analysis is peripheral blood, since samples from neural tissues, which are directly related to NDDs, are not easily accessible. Methylation alterations initiated at embryonic and fetal stages of pattern formation and development are typically consistent throughout development and are detectable in many tissues, including peripheral blood [54]. However, if methylation alterations evolve later in development, they might cause mosaicism and, therefore, may not appear in blood but may appear in other tissues. For instance, in three patients with Beckwith-Wiedemann syndrome (BWS), the methylation levels in blood were normal, while tongue tissue samples showed an episignature [55].

### 2.2. Requirement for Large Study Cohorts and Reference Databases

The utility of episignatures as a reliable diagnostic tool in clinical settings requires access to large reference databases and large study cohorts of the constitutional condition under study with individuals that are clinically and genetically diagnosed with the syndrome. Larger cohorts of patients increase the accuracy in machine learning algorithms and, therefore, result in more definitive predictions. Due to the rarity of NDDs/OGIDs, and the challenges in their clinical and genetic diagnosis, the related databases are typically very small. This sometimes contributes to the failure of the signature detection process, such as in the case of Weaver syndrome (WVS) [4]. However, combining the WVS and Cohen-Gibson syndrome (COGIS), which share a common genetic etiology, and thereby increasing the size of the reference cohort, has now enabled the discovery of a unique episignature (unpublished results). In another study, increasing the size of the patient cohort helped identify a signature specific to WVS [56]. Similarly, no episignature was observed previously for Coffin-Siris syndrome (CSS) [4], but increasing the number of cases and adding patients with Nicolaides-Baraitser syndrome (NCBRS) resulted in the identification of a DNA methylation pattern shared between the two disorders [43].

### 2.3. Various Types of Episignatures

Most syndromes caused by variants in a single gene are associated with one episignature specific to that disorder [4,39,40,46,48,50,57,58,59,60]. Sometimes, however, there is an overlap between the episignatures of two syndromes. For example, CHARGE and Kabuki syndromes have overlapping clinical presentations and show similar hypermethylation levels at probes on *HOXA5* [46]. Bjornsson et al. attributed the significant phenotypic overlap in these disorders to the overlapping functional roles of genes involved in epigenetic regulatory machinery [61]. On one extreme, identical DNA methylation signatures are observed throughout multiple genes of the same protein complexes. Coffin-Siris syndrome (CSS), Nicolaides–Baraitser syndrome (NCBRS), and Chr6q25 microdeletion syndrome, for instance, belong to BAFopathies or SWI/SNF remodeling complex disorders and share an episignature [43]. Notably, some subtypes of CSS and NCBRS have a higher overlap between methylation alterations than within CSS [43]. CdLS types 1–4, as well as Kabuki syndrome types 1 and 2, are other syndromes that present an identical episignature, despite different causal genes [35].

On the other extreme, some single-gene syndromes are associated with distinct episignatures, based on the location of the mutation on the underlying gene. For instance, ADNP patients with variants inside c. 2000–2340 (ADNP central—ADNP_C) present a different DNA methylation pattern compared to those with variants outside that region (ADNP terminal—ADNP_T) because of the difference in the associated protein domains, while manifesting similar clinical features [62]. In contrast, mutations in *KAT6B* result in two distinct episignatures associated with two different syndromes—namely, Genitopatellar (GTPTS) and Say-Barber-Biesecker-Young-Simpson syndromes (SBBYSS) [35].

Another interesting reported fact is that a linear relationship is seen between the dosage of the defective protein and the intensity of DNA methylation alterations in some syndromes, such as immunodeficiency-centromeric instability-facial anomalies syndrome types 2–4 (ICF2–4) associated with mutations in the zinc finger and BTB domains [35,63].

## 3. DNA Methylation Analysis and Classification Models

For a summary of this section, see Figure 1.

### 3.1. Methylation Assessment

In order to assess the methylated and unmethylated signal intensities, DNA samples typically collected from peripheral blood are applied to Illumina Infinium Human Methylation27 BeadChip [33,36], Illumina Infinium HumanMethylation450 BeadChip [4,35,36,39,40,41,43,45,46,48,49,50,51,57,62,64], or an Illumina Infinium MethylationEPIC kit [4,35,41,43,51,52,62] after bisulfite conversion. Methylation levels at each probe, or β-values, are measured as the ratio of the methylated probe signal to the sum of the methylated and unmethylated probe signal intensities. Probes overlapping single-nucleotide polymorphisms (SNP) and X or Y chromosomes are normally excluded [41].

### 3.2. Probe Selection

Our previous studies have shown that this method functions best while using ~150 markers [35]. The DNA methylation profile at the selected CpG sites is called the episignature, which is then implemented for the construction of the classification model. If the methylation data for all CpG sites are used for building the model, it increases the chance of model overfitting—a condition in which the model performs perfectly on the training set but cannot classify samples in the test set correctly [65]. Therefore, dimension reduction or a selection of the subset of probes is done to avoid overfitting. The other reason behind probe selection is to ensure that the model is as simple as possible, using only the probes that make the most contribution to the disease under investigation [65]. Typically, a CpG site is considered differentially methylated if there is a minimum of a 5–20 percent difference in the average methylation level of that site between the case subjects and controls, with a corrected *p*-value < 0.01. In the absence of a sufficiently large number of differentially methylated CpGs, we identify the syndrome as not bearing an episignature or having a milder signature below a sensitive detection capacity. Probe selection is done by implementing multivariable linear regression (MLR) modeling [4,35,41,43,46,49,50,51,62], analysis of variance (ANOVA) [40,45,48,57], and/or a Mann-Whitney U test [36,39,46]. We should remark that both ANOVA and linear regression should be performed on *M*-values, derived by applying logit transformation on the *β*-values in order to obtain a normal distribution and homoscedasticity (equal variance across all probes), which are the requirements of these statistical tests [66]. A Mann-Whitney U test, in contrast, does not assume a normal distribution of the data and is more robust against heteroscedasticity [67]. Nevertheless, the use of *M*-values is recommended for performing a differential methylation analysis, since it guarantees a higher true positive rate (TPR) [66,67]. In some studies, probe selection has involved two more steps—namely, a receiver operating characteristic (ROC) curve analysis and elimination of highly correlated probes [35,41].

### 3.3. Unsupervised Methods and Signature Assessment

An unsupervised machine learning model is usually used to test the strength of probes in differentiating cases from either controls or other syndromes. An unsupervised model is a model utilized for clustering and/or dimension reduction in which the subjects’ labels, i.e., case and control, are either not available or ignored, and the model divides the subjects into clusters based only on their methylation data. Researchers usually perform a few various techniques, including *k*-means clustering [36,43], *k*-median clustering [36], hierarchical clustering [4,35,36,39,40,41,43,46,48,49,50,51,52,62], principal component analysis (PCA) [36], multiple dimensional scaling (MDS) [35,51,52,62], and t-distributed stochastic neighbor embedding (t-SNE) [35,41], followed by creating a graph based on the model to visualize the resulting clusters in order to verify the robustness of the signature.

### 3.4. Supervised Classification Models

Unsupervised models often create diagrams with some overlap between patients and healthy controls or between cases of different diseases. In order to separate cases and controls with a high level of confidence, usually a supervised classification model is also constructed. The most popular classifiers implemented for the purpose of differentiating cases and controls based on methylation data are support vector machines (SVM) and random forests (RF) [4,35,41,43,46,49,50,62,68,69,70]. The rationale behind the selection of these methods is their capability in processing data with low numbers of data points (samples) and high dimensionality, i.e., a large number of probes (CpGs), which are the typical characteristics of genomic datasets and DNA methylation datasets in particular.

SVM are supervised algorithms suitable for two-group classification problems. Conceptually, SVM separates the two groups by constructing a hyperplane with one dimension less than the number of probes [71]. For instance, if the data has two probes, the two groups are separated with a line, and if there are three probes, the partition is performed by a two-dimensional plane. This hyperplane is called a linear kernel. The hyperplane defines an optimal margin keeping a maximal distance from the data points, ensuring an explicit partition between the groups. If the two groups are not linearly separable, the data is transformed to a higher dimension by applying another kernel function, such as a polynomial kernel and radial basis function (RBF) [71].

There are two main problems when handling microarray data—namely, computational complexity and overfitting [72]. SVM can resolve both issues. They reduce the computational complexity by a method called the kernel trick. This allows computations in a lower dimension, negating the need to transform the data to a higher dimension [71]. SVM are robust against overfitting, since they do not learn based on the whole training set, but they only use those data points that fall within the margin, called the support vectors [72]. Therefore, the hyperplane does not move on the addition of new data points unless they are detected as support vectors. Modifications in the model allowed for a few misclassifications in the training set, as far as they fall within the margin, which further helped the algorithm to avoid overfitting [72].

In their classic form, SVM are nonprobabilistic models classifying new data points with one of the two groups without providing the probability that the data point belongs to a group. Platt modified the method, enabling it to perform as a probabilistic one whenever required, using the distance between each point to the hyperplane to compute the point’s score. This score is a number between 0 and 1 and it is the probability of that point belonging to a group [73]. Regarding the problem of classifying cases and controls based on their methylation data, this score is called the methylation variant pathogenicity (MVP) score, and it is the probability of a patient belonging to the disease class [73]. Subjects with scores normally above 0.5 are classified as having a methylation profile related to the syndrome under investigation. SVM has been applied as a classifier in many genome-wide DNA methylation-based diagnoses of Mendelian neurodevelopmental disorders [4,35,41,43,46,49,50,62].

Random forest (RF) is another convenient model applied as a classifier [51]. Random forest is an ensemble machine learning model comprised of many decision trees. Each tree is built by a random selection of samples and probes. The randomness in the construction of decision trees maximizes the model’s accuracy. Moreover, random forests do not overfit, since they provide the result by averaging over numerous decision trees [74]. By considering the number of decision trees voting for each class, the model computes a confidence score between 0 and 1 for each case subject, with a score higher than 0.5 meaning the subject is classified with the case group [51]. RF are not as popular as SVM for the classification of patients with various neurodevelopmental disorders; however, their performance on DNA methylation data appears to be promising, since they have been utilized in several studies investigating the relationship between DNA methylation and other diseases, including various types of cancers [68,69,70]. One reason for the lower popularity of RF in comparison with SVM may be the fact that RF is more prone to the choice of the hyperparameters, while SVM demonstrates negligible changes when different hyperparameters are utilized [75].

## 4. Resolving the Unresolved

Variants of unknown significance are a common result of gene, whole-exome, and genome sequencing. This becomes more challenging in rare genetic conditions with ambiguous clinical presentations. With the implication of the supervised machine learning classifier, trained for the classification of subjects of many syndromes simultaneously, we can provide a diagnosis for many unresolved cases, thereby enabling clinical reclassification of VUSs [4,35,41,43,46,49,51,52,62]. Besides classifying patients with VUSs, this multi-class model has further been utilized to reclassify patients with an incorrect diagnosis. For instance, Figure 2A illustrates the MVP scores generated by a multi-class SVM (a collection of many two-class SVMs), i.e., a combination of several SVMs, each trained by comparing the methylation data of individuals with one syndrome against individuals from controls and 14 other syndromes. Six unresolved cases suspected of having MRXSCJ were supplied into the model. It can be observed that four of them were classified as MRXSCJ cases, one as not having any of the 15 syndromes, and one with a Sotos1 syndrome episignature [41]. Figure 2B depicts the corresponding MDS plot, where MRXSCJ samples, Sotos1 samples, and healthy control samples are perfectly separated from each other. Four of the suspected MRXSCJ cases are clustered with the other MRXSCJ samples, one with controls—hence, remaining unresolved—and one with Sotos1 samples.

## 5. Future Perspectives

### 5.1. Implementation of Other Probe Selection and Classification Models

The choice of probe selection and classification models plays an essential role in the correct detection of the episignature and definitive classification of the patients. Therefore, it is extremely important to ensure that the selected models are the most appropriate for our data. The common practice in detecting the DNA methylation signature of Mendelian neurodevelopmental syndromes has been to apply various probe selection methods in several steps, eliminating a portion of nondifferentially methylated CpG sites at a time, and to construct a supervised classification model based on the selected probes. Although this method has proven successful in classifying the unresolved cases for numerous congenital anomalies, there are other approaches that can be explored, possibly creating more promising results. SVM is the classification model that has been utilized the most for diagnosing developmental disorders. There is a probe selection method called recursive feature elimination (RFE) that utilizes SVM and has been established as a powerful probe selection tool, together with an SVM classifier for microarray data [65]. Moreover, the radial basis kernel has been suggested as the kernel that provides the highest performance with expression data [76].

### 5.2. Evolving Episignatures—EpiSign Knowledge Database (EKD)

Future use of epigenomics in the screening and classification of rare disorders highly depends on the availability of a large reference DNA methylation episignatures database. Establishing a robust database with epigenetic profiles from thousands of patients and control individuals can guide us to uncover episignatures in many rare genetic conditions that are currently categorized as episignature-negative conditions. Our initial understanding about known gene/disease-specific peripheral blood episignatures came from small case-control analyses, but the subsequent implementation of a genomic DNA methylation analysis for the clinical screening of patients with DD/ID has helped us create an EpiSign Knowledge Database (EKD). The expansion of this database is underway, with the implementation of large-scale clinical trials of this technology, like the recently announced Canadian national trial EpiSign-CAN (https://www.genomecanada.ca/en/beyond-genomics-assessing-improvement-diagnosis-rare-diseases-using-clinical-epigenomics-canada).

The continued development of computational models we described in this review, coupled with advancements in technology and an increasing understanding of disease-specific episignatures, will expand the breadth of diseases, including NDDs, that can be diagnosed by epigenetic assays.

## Figures and Tables

**Figure 1 ijms-21-09303-f001:**
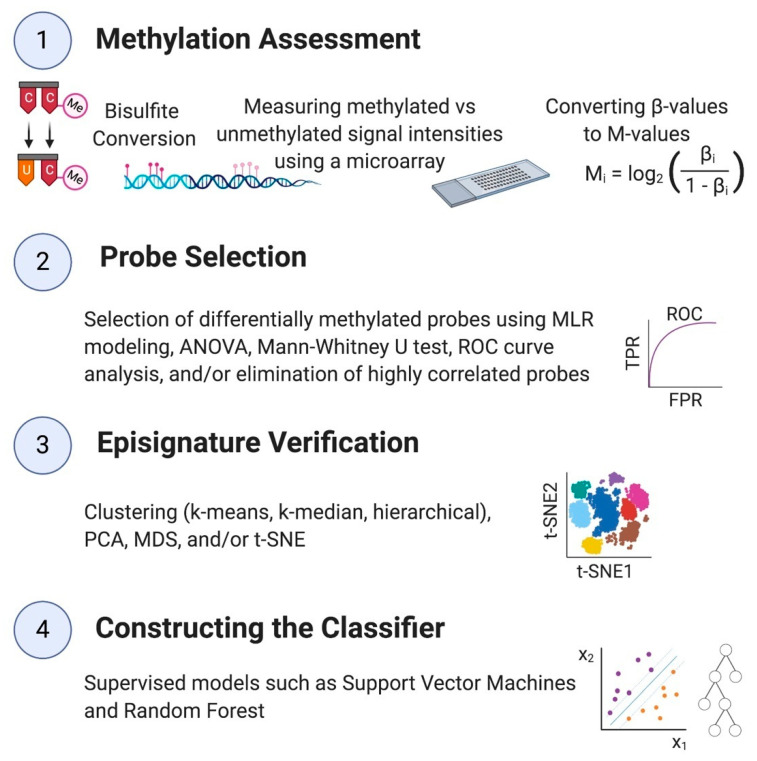
Summary of a DNA methylation analysis and constructing the classification model.

**Figure 2 ijms-21-09303-f002:**
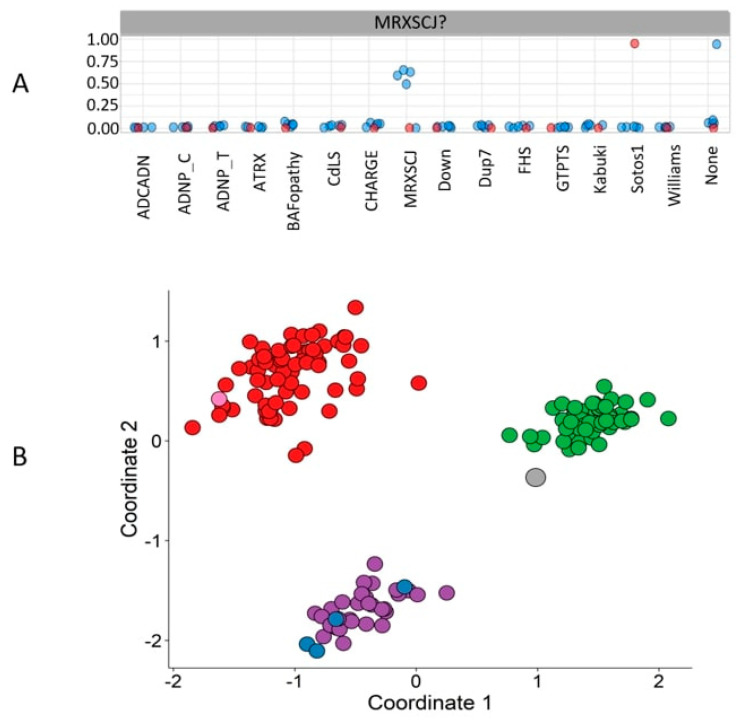
Classification of unresolved cases. (**A**) Six individuals suspected of having mental retardation, X-linked syndromic, Claes-Jensen-type (MRXSCJ) were supplied to the multi-class support vector machine (SVM) classifier. The methylation variant pathogenicity (MVP) scores generated by the model illustrate that 4 individuals are classified as cases of MRXSCJ, one as not having any of the 15 syndromes, and one (the red one) is classified as a Sotos syndrome 1 (Sotos1) case. (**B**) The multiple dimensional scaling (MDS) plot with MRXSCJ samples (purple circles), healthy control samples (green circles), and Sotos1 samples (red circles). This plot illustrates the clustering of four of the individuals described in panel A with the MRXSCJ case samples (blue circles) and two individuals separate from the MRXSCJ case samples. The grey circle represents the patient classified as not having any of the 15 syndromes, and the pink circle depicts the individual classified as a Sotos1 case.

**Table 1 ijms-21-09303-t001:** List of neurodevelopmental syndromes with a detected episignature.

Neurodevelopmental Disorder		Abbreviation	Underlying Gene(s)/Locus	Phenotype MIM Number	Episignature Published
Alpha-thalassemia mental retardation syndrome		ATRX	*ATRX*	301040	Yes [4,35,40,41]
Arboleda-Tham syndrome (formerly MRD32)		MRD32	*KAT6A*	616268	No
Autism, susceptibility to, 18		AUTS18	*CHD8*	615032	Yes [35,42]
BAFopathies	Coffin-Siris 1–4 (CSS1–4)	BAFopathy	*ARID1A*, *ARID1B*, *SMARCB1*, *SMARCA4*	135900, 614607, 614608, 614609	Yes [35,41,43]
	Nicolaides-Baraitser (NCBRS) syndromes		*SMARCA2*, *SMARCC2*	601358	
Beck-Fahrner syndrome		BEFAHRS	*TET3*	618798	No
Blepharophimosis intellectual disability SMARCA2 syndrome		BISS	*SMARCA2*	N/A	Yes [44]
Börjeson-Forssman-Lehmann syndrome		BFLS	*PHF6*	301900	Yes [35]
Cerebellar ataxia, deafness, and narcolepsy, autosomal dominant		ADCADN	*DNMT1*	604121	Yes [4,35,41,45]
CHARGE syndrome		CHARGE	*CHD7*	214800	Yes [4,35,41,46]
Chr16p11.2 deletion syndrome		Chr16p11.2del	Chr16p11.2del	611913	Yes
Cohen-Gibson syndrome (PRC2 complex, shares signature with Weaver syndrome)		COGIS	*EED*	617561	No
Cornelia de Lange syndrome 1–4		CdLS1–4	*NIPBL, RAD21, SMC3, SMC1A*	122470, 300590, 610759, 614701	Yes [35,41]
Down syndrome		Down	Chr21 trisomy	190685	Yes [35,41,47]
Dystonia-28, childhood-onset		DYT28	*KMT2B*	617284	No
Epileptic encephalopathy, childhood-onset		EEOC	*CHD2*	615369	Yes [35]
Floating Harbor syndrome		FLHS	*SRCAP*	136140	Yes [4,35,41,48]
Genitopatellar syndrome		GTPTS	*KAT6B*	606170	Yes [4,35,41]
Helsmoortel-Van Der Aa syndrome (ADNP syndrome (Central))		HVDAS_C	*ADNP*	615873	Yes [35,41]
Helsmoortel-Van Der Aa syndrome (ADNP syndrome (Terminal))		HVDAS_T			
Hunter McAlpine syndrome		HMA	Chr5q35-qter duplication involving *NSD1*	601379	Yes [35]
Immunodeficiency-centromeric instability-facial anomalies syndrome 1–4		ICF_1–4	*DNMT3B, ZBTB24, CDCA7, HELLS*	242860, 614069, 616910, 616911	Yes [35]
Intellectual developmental disorder, X-linked, syndromic, Armfield-type		MRXSA	*FAM50A*	300261	No
Kabuki syndrome 1 and 2		Kabuki	*KMT2D, KDM6A*	147920, 300867	Yes [4,35,41,46,49]
Kleefstra syndrome 1		Kleefstra	*EHMT1*	610253	Yes [35]
Koolen-de Vries syndrome		KDVS	*KANSL1*	610443	Yes [35]
Mental retardation, autosomal dominant 23		MRD23	*SETD5*	615761	No
Mental retardation, autosomal dominant 51		MRD51	*KMT5B*	617788	Yes [35]
Mental retardation, X-linked 93		MRX93	*BRWD3*	300659	Yes [35]
Mental retardation, X-linked 97		MRX97	*ZNF711*	300803	Yes [35]
Mental retardation, X-linked syndromic, Claes-Jensen-type		MRXSCJ	*KDM5C*	300534	Yes [4,35,41,50]
Mental retardation, X-linked syndromic, Nascimento-type		MRXSN	*UBE2A*	300860	Yes [35]
Mental retardation, X-linked, Snyder-Robinson-type		MRXSSR	*SMS*	309583	Yes [35]
Ohdo syndrome, SBBYS variant		SBBYS	*KAT6B*	603736	Yes [4,35]
Phelan-McDermid syndrome		PHMDS	*SHANK3*	606232	No
Rahman syndrome		RMNS	*HIST1H1E*	617537	Yes [35,51]
Rubinstein-Taybi syndrome		RSTS	*CREBBP, EP300*	180849, 613684	Yes [35]
SETD1B-related syndrome		SETD1B	*SETD1B*	N/A	Yes [52]
Sotos syndrome 1		Sotos1	*NSD1*	117550	Yes [4,35,39,41]
Tatton-Brown-Rahman syndrome		TBRS	*DNMT3A*	615879	Yes [35]
Weaver syndrome (PRC2 complex, shares signature with Cohen-Gibson syndrome)		WVS	*EZH2*	277590	No
Wiedemann-Steiner syndrome		WDSTS	*KMT2A*	605130	Yes [35]
Williams-Beuren region duplication syndrome (Chr7q11.23 duplication syndrome)		Dup7	Chr7q11.23 duplication	609757	Yes [35,41,53]
Williams-Beuren syndrome (Chr7q11.23 deletion syndrome)		Williams	Chr7q11.23 deletion	194050	Yes [35,41,53]
Wolf-Hirschhorn syndrome		WHS	Chr4p16.3 deletion	194190	No

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
