# Peer review of "Diagnostic Utility of Genome-Wide DNA Methylation Analysis in Mendelian Neurodevelopmental Disorders"

_ijms, 2020, doi:10.3390/ijms21239303_

Round 1

Reviewer 1 Report

The authors made a comprehensive and interesting review in one of the topic of the moment in the mendelian genetics. 

The paper is well written, readable, and finally a good paper to be accepted. 

Reviewer 2 Report

In the manuscript titled  “Diagnostic Utility of Genome-wide DNA 2 Methylation Analysis in Mendelian 3 Neurodevelopmental Disorders” the authors summarize the knowledge of the DNA methylation episignatures associated with rare disorders. They cover most of the topics including genetics testing model, syndromes, and computational analysis. They also make several recommendations and  postulate future perspectives

The manuscript is well done and only minor considerations should be taken into account.

  • Most of the manuscript focuses on the use of the arrays, although the NGS is been used to analyze methylation, then it should be useful to include the contribution of this new technology and try to compare both approaches.
  • Line 82 the references could be actualized
  • Line 110 perhaps, the expression early report is not correct because they are references of 2010.
  • Line 190 there is a mistake
  • Line 198 a reference need to be added
  • I have a question: how many patients and controls are necessary to perform a methylation analysis?
